# Parameterized Anchor Representations via Adaptive Matrices (PARAM) for Relative-Representations

Oscar Thorsted Svendsen*1, Nikolaj Holst Jakobsen*1, Hiba Nassar†1, and Fabian Mager†1

$^1 Technical University of Denmark$
{s224177,s234818,hibna,fmager}@dtu.dk

## 1    Introduction

In 2023 Moschella et al.[1] introduced Relative Representations (RR) as the first framework enabling zero-shot stitching of neural components, showing that latent spaces from independently trained models can communicate through relative similarity to shared anchors due to similar internal representations[2]. Building on this foundation, we aim to improve its robustness and practical applicability. Since data points are defined through their relations to anchors, the quality of RRs depend critically on how well the anchors describe the latent manifold. Each anchor must be situated in a semantic context that captures both broad dissimilarities and fine-grained local relations. For instance, the embedding of *"King"* should be close to *"Queen"* and *"Castle"* but distant from *"Banana"* and *"Space"*. Hence, the anchor set should comprehensively cover the latent space while representing well-defined prototypes of their semantic areas for the sake of robustness across spaces.

In the original formulation, anchors were randomly chosen, and the RR was calculated using cosine similarity. While enough random points tend to cover a space, they provide no guarantee of optimal coverage or anchor robustness.

Moreover, cosine similarity, though efficient in high-dimensional settings, discards vector norms, which can encode information such as feature confidence[3], and remains sensitive to translations in the embedding space - especially near the origin where embeddings tend to concentrate[4].

## 2    Method

### 2.1    Learning anchors as mixtures (PARAM)

We parameterize each anchor as a convex mixture of examples from a shared subset. Let $X_{\text{sub}} \in \mathbb{R}^{k \times d}$ denote the subset (rows are embeddings) and let $P \in \mathbb{R}^{m \times k}$ be a row-stochastic weight matrix (im-

plemented via a row-softmax). An anchor set is

$$A = P X_{\text{sub}}, \qquad a_i = \sum_{j=1}^{k} p_{ij} x_j,$$

which provides flexible, denoised placements. Given multiple encoder banks $\{X^{(b)}\}$, we reuse the same $P$ to form bank-specific anchors $A^{(b)} = P X_{\text{sub}}^{(b)}$; RR features are then computed per bank with a differentiable similarity. Pairwise alignment losses between banks act on their RR features, and gradients flow through the similarity into $P$.

### 2.2    Whitened Inner Product (WIP)

A suitable RR similarity should be robust to translation, rotation, and anisotropic scaling across embedding spaces, and must be differentiable to train $P$. The similarity used in this paper is a *whitened inner product*.

Let $\mu$ be the empirical mean and $\hat{\Sigma}$ the empirical covariance of the target bank. We use a shrinkage covariance

$$\Sigma_\lambda = (1-\lambda)\,\hat{\Sigma} + \lambda\,\frac{\text{tr}(\hat{\Sigma})}{d}\,I + \varepsilon I, \qquad L = \Sigma_\lambda^{-1/2}.$$

For an embedding $x$ and anchor $a_i$, the WIP score is

$$s_i^{\text{WIP}}(x) = \langle (x-\mu)L,\ (a_i-\mu)L \rangle = (x-\mu)^\top \Sigma_\lambda^{-1} (a_i - \mu).$$

WIP is translation-invariant (via $x - \mu$), anisotropic-scale–invariant (via whitening by $L$), and rotation-invariant (inner product in the whitened space). By retaining this magnitude information while accounting for local covariance structure, WIP captures both geometric orientation and local variance - yielding more robust relative representations.

### Training losses

On top of WIP RR features, we combine three alignment objectives across encoder banks: (i) *Soft Jaccard* loss promotes fine-grained similarity inside clusters by encouraging similar anchor-wise activations across encoders; (ii) *Barlow Twins* [5] enforces invariance to encoder-specific distortions while reducing redundancy between RR dimensions; and (iii)

---

*These authors contributed equally to this work and share first authorship.

†These authors advised the project.

*Alignment–Uniformity* [6] pulls corresponding points closer in RR space without collapsing all features into a degenerate solution.

These losses act in complementary ways: Soft Jaccard preserves local similarity patterns, Barlow Twins stabilizes global feature structure, and Alignment–Uniformity balances clustering with diversity. We further regularize anchors with an *anchor cohesion* loss, which biases each anchor to be explained primarily by nearby points (in the whitened space), reducing noise from distant, unrelated samples. The result is anchors that cover the space while remaining well localized and stable across embedding variations. Expressing each anchor as a weighted combination of its closest datapoints improves robustness to encoder-specific shifts and enhances generalization to unseen spaces.

## 3    Experiments and Results

The method is evaluated in a zero-shot image classification setting following the Moschella et al. (2023) framework.

Each zero-shot configuration consists of frozen encoders that generate absolute embeddings. These embeddings are subsequently projected into their respective relative representation spaces using a shared set of parallel anchors. A relative decoder is trained on one of these spaces, and since the decoder operates within the shared relative representation domain, embeddings from the remaining encoders can be directly employed within the same classifier - thereby enabling zero-shot transfer across encoders. For generalization purposes, the encoder that is zero-shotted on the relative decoder is excluded when training the PARAM anchors.

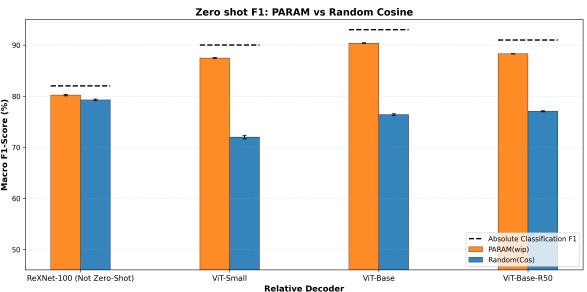

**Figure 1.** F1 score of zero-shot classification of relative embeddings from a pretrained *RexNet-100* encoder. The classifiers are trained on relative embeddings from pretrained *ViT-small, ViT-Base,* and *ViT-Base-R50* encoders.

The results are directly comparing the method described in Moschella et al. (2023) to the proposed PARAM method.

## 4    Discussion and Limitations

Figure 1 shows that the proposed PARAM relative representation method can substantially increase zero-shot classification performance compared to using random anchors and cosine similarity. Similar performance increases were observed in preliminary cross-lingual and other zero-shot experiments.

A noteworthy observation is that this is the first empirical demonstration of zero-shot performance outperforming a same-architecture absolute classifier trained on the regular encoder (the dotted line above RexNet-100).

This shows that instead of training a classifier directly on a simple encoder, it is possible to improve classification performance by zero-shotting to a classifier trained on higher-quality embeddings. In addition, the small gaps from the zero-shot performance to the absolute performance indicate that high-quality embeddings are mostly important for training a competent classifier. Then, when it comes to using the classifier, the results show that, in some cases, using a lighter encoder can mimic close to a similar performance.

This opens up the possibility of training a single high-quality classifier on relative representations derived from a strong encoder and subsequently deploying it at inference across a range of lighter or task-specific encoders without retraining. Such a setup could significantly reduce training and maintenance costs while retaining much of the performance of large models. The setup highlights the potential of Relative Representations not only for zero-shot transfer but also as a means of decoupling encoder complexity from downstream classifier performance.

There are a couple of things to consider when implementing relative representations that were discovered in this research process. In a few settings, such as cross-lingual transfer, parallel points must be created manually as opposed to the CIFAR experiment. Preliminary research hints at a positive correlation between the number of parallel points and zero-shot performance.

Moreover, while fully parallel data provides the most stable alignment, preliminary observations suggest that fully parallel points might not be needed. We are researching partial or class-level parallel sets that could contribute meaningfully to anchor consistency. This indicates that relative representations could extend beyond strictly parallel datasets - potentially allowing zero-shot stitching even in settings where zero parallel points are available.

Moschella et al. (2023) demonstrated that relative representations enable zero-shot stitching of neural components. This work extends the foundation with a robust relative space and the ability to decouple encoder complexity from downstream inference.

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
