# OpenReview forum: "Parameterized Anchor Representations via Adaptive Matrices (PARAM) for Relative-Representations"
_NLDL.org/2026/Abstracts_Track — NLDL 2026 Abstracts_

### Official Review · Reviewer_4XKx · 2025-10-24

**Soundness:** 3
**Correctness:** 3
**Rating:** 4
**Confidence:** 4

**Summary:**

The paper builds on an existing framework called relative representations, and proposes some new ideas with the aim of improving the robustness and practical utility of the framework. The framework enables zero-shot stitching of neural components by aligning latent spaces of independently trained models through shared anchor similarities.

**Strengths:**

The novelty of the paper is two-fold: 1) It proposes a method for learning anchors from mixtures instead of choosing them randomly and 2) it proposes a new measure for similarity, namely the whitened inner product, which is robust to translation, rotation, and anisotropic scaling across embedding spaces. These ideas are innovative and the experiments verify that the performance improves in a zero-shot image classification setting by incorporating them into the framework.

**Weaknesses:**

More experiments and use cases should be added to show the robustness of the new modified framework.

---

### Official Review · Reviewer_4aTv · 2025-10-31

**Soundness:** 3
**Correctness:** 3
**Rating:** 4
**Confidence:** 3

**Summary:**

The authors introduces PARAM (Parameterized Anchor Representations via Adaptive Matrices), an improved framework for Relative Representations (RR) that enhances robustness and practical applicability in zero-shot neural component stitching.  The key contributions include adaptive anchors parameterized as convex mixtures for better latent space coverage, a Whitened Inner Product (WIP) similarity measure for robust relative representations, and novel training losses to optimize alignment, stability, and diversity. Experiments show that PARAM significantly improves zero-shot classification performance compared to random anchors and cosine similarity, even outperforming classifiers trained on absolute embeddings.  This approach enables the decoupling of encoder complexity from downstream classifier performance, reducing training costs while maintaining high accuracy.

**Strengths:**

The paper’s key strengths include its novel adaptive anchors, which use convex mixtures to ensure robust and well-covered latent space representations, and the introduction of the Whitened Inner Product, a similarity measure that preserves magnitude and local covariance while being robust to translation, rotation, and scaling. Its strong zero-shot evaluation demonstrates that PARAM outperforms the original RR framework and even absolute classifiers trained on encoder embeddings. Additionally, the method enables a single high-quality classifier to be deployed across multiple encoders without retraining. Overall, the paper is well written, clearly organized, and presents its contributions in a concise and accessible manner.

**Weaknesses:**

While the paper presents strong conceptual contributions, it lacks sufficient implementation details, such as hyperparameters, training configurations, and comprehensive information about the datasets used, including their size, characteristics, and preprocessing steps. Additionally, although the combination of training losses (e.g., Soft Jaccard, Barlow Twins, Alignment–Uniformity) is discussed, the paper does not provide detailed ablation studies to quantify the individual contribution of each loss to the overall performance.   We understand that there are page constraints that may limit the inclusion of all these details in the paper. However, we encourage the authors to address some of these points on the poster.

---

### Official Review · Reviewer_TqGA · 2025-11-03

**Soundness:** 2
**Correctness:** 2
**Rating:** 4
**Confidence:** 4

**Summary:**

This extended abstract is about relative representations (RRs). The idea is to transform a latent space from an absolute coordinate system to a relative coordinate system, by representing each sample based on its distance and/or similarity to a fixed set of anchor samples. This yields a unified, shared latent space and enables zero-shot communication and model stitching between different networks without additional training.

The authors argue that this approach has shortcomings: (i) anchors are chosen randomly, (ii) the cosine similarity does not take vector norms into account and is sensitive to translations. As a remedy, the authors parametrize anchors as convex combinations of embeddings (allows for learning the anchors) and use a whitened inner product to score data points. Together with a specific choice of training losses and regularizers, the authors demonstrate promising preliminary results on zero-shot classification.

**Strengths:**

- The use of relative representations is appealing.
- Representation learning and zero-shot classification is highly relevant for this conference.
- Their approach is motivated.
- The preliminary results appear to be promising.

**Weaknesses:**

- The paper lacks clarity and is thus (in parts) not easily readable. For example, the reader has to guess and/or infer the meaning of some concepts and variables from context.
- Essential parts of the method and the experiments are either not at all or insufficiently provided.
- The paper is not self-contained as it cannot be understood without reading [1]. This can be seen by the usage of terminology in Section 1 and the experimental setting in Section 3.


More comments:
- Some concepts could be introduced better. I.e., for readers unfamiliar with, e.g., [1], the terms *anchor* and *bank* could be introduced better to make the paper more accessible.
- After line 032, I was expecting at least one sentence stating the idea of the paper.
- Lines 036-040: What is a *shared subset*? The *subset* is a subset of what set? Where do $X$ and $P$ come from? The variables $k$, $d$, and $m$ could be introduced in a better way. How is that *denoised*? What is $b$?
- Line 058: The reader has to guess/assume that $I$ is the identity matrix, $d$ is the dimensionality of $\mu$, $\lambda \in [0,1]$ and $\epsilon \geq 0$, but it would be better if the reader does not need to guess/assume that. What is a *target bank*?
- Why are the training losses and regularizers not stated as equations?


Minor comments:
- affiliation: broken style
- some spaces are missing, e.g., lines 008, 029, 032, etc.
- inconsistent use of dashes, e.g., lines 031, 063, 067, etc.
- space violation in line 061
- citation styles mixed, e.g., lines 002 and 096
- mixed style in reference list, e.g., compare lines 171 and 176

---

### Decision · Program_Chairs · 2025-11-05

**Decision:**

Accept

**Comment:**

The abstract is of interest to the community and should be presented at the conference.